# MAAT: Multi-timestep Alternating Adversarial Training Against Personalized Content Security in Diffusion Models

## Abstract

Despite the remarkable progress in fine-tuning text-to-image diffusion models for personalized content generation, these techniques pose serious societal risks when misused for generating fake news or malicious individual-targeted content. Existing anti-customization strategies predominantly rely on adversarial attacks. However, their effectiveness remains limited due to inadequate exploration of the intrinsic properties of diffusion models. In this paper, we propose **M**ulti-timestep **A**lternating **A**dversarial **T**raining (**MAAT**), a novel approach that disrupts unauthorized model customization by strategically intervening in the diffusion process. MAAT formulates adversarial attacks as a sparse multi-task optimization problem over diffusion timesteps, and introduces an **A**daptive **N**on-uniform **T**imestep **G**radient **E**nsemble (**ANTGE**) to efficiently select representative timesteps. This approach enhances attack performance while significantly reducing computational overhead. We further propose a **L**ayer-**A**ware **A**ttention **T**argeting (**LAT**) loss, which jointly disrupts self-attention and cross-attention modules by selectively targeting layers whose attention maps highly correlate with identity-related regions such as faces. In addition, MAAT is a two-stage training paradigm, incorporating surrogate model pre-training and iterative adversarial refinement. Extensive experiments on two benchmark facial datasets validate that MAAT significantly outperforms existing methods in various white-box and black-box attack scenarios, with more than 20% improvements in ISM and 6.5% improvements in FDFR. The source code is available at https://anonymous.4open.science/r/MAAT-C450.

## 1 Introduction

In recent years, diffusion models (Ho et al., 2020; Song et al., 2020; Rombach et al., 2022; Dhariwal & Nichol, 2021) have made remarkable progress in various tasks, demonstrating exceptional capabilities in high-fidelity image synthesis (Ho et al., 2020; Song et al., 2020; Dhariwal & Nichol, 2021), precise image editing (Meng et al., 2021; Huang et al., 2025), text-to-image generation (Nichol et al., 2021; Ramesh et al., 2022; Saharia et al., 2022; Rombach et al., 2022), and video synthesis (Li et al., 2024b; 2025; Wang et al., 2025). Among these advancements, Personalized Content Synthesis (PCS) has emerged as a particularly prominent application (Gal et al., 2022; Ruiz et al., 2023; Kumari et al., 2023). While PCS enables the generation of high-quality and diverse visual content, its customization capabilities also open the door to malicious misuse, particularly in facial forgery. As shown in Figure 1(a), users can leverage PCS techniques DreamBooth (Ruiz et al., 2023) to create fake photorealistic facial images. Such forgeries pose serious risks to individual privacy and reputation. In extreme cases, these forgeries may mislead public opinion or trigger political unrest. Thus, it is essential to develop effective measures to safeguard users against these malicious applications.

To this end, recent research has developed proactive protection methods that introduce imperceptible perturbations onto images, aiming to disrupt the outputs of diffusion models. Although contemporary methods demonstrate limited effectiveness in countering unauthorized personalization attacks on Stable Diffusion, four fundamental limitations hinder their practical deployment. First, existing methods neglect nuanced internal properties intrinsic to the diffusion model, either adopting strategies detached from the fundamentals of the diffusion model (Rombach et al., 2022; Shan et al., 2023) or narrowly focusing on adversarially maximizing reconstruction loss (Liang et al., 2023; Liang &

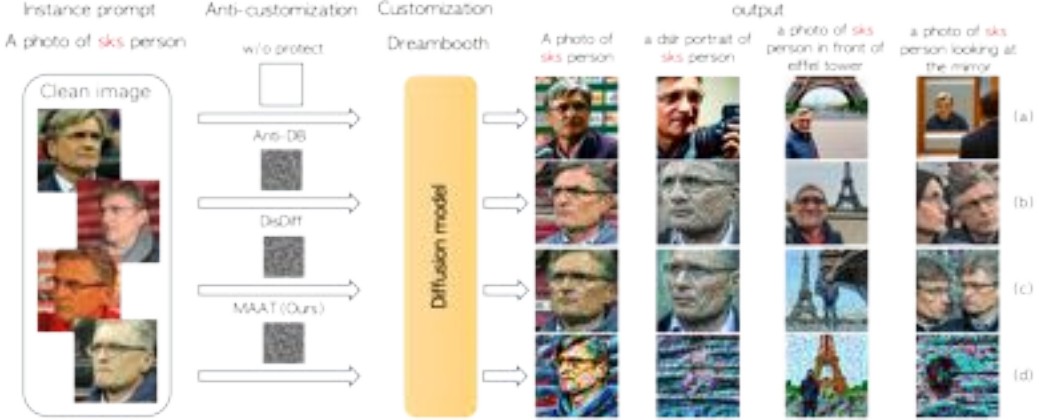

Figure 1: Comparison of DreamBooth (Ruiz et al., 2023), Anti-DreamBooth (Van Le et al., 2023), DisDiff (Liu et al., 2024) and the proposed MAAT. The first row shows the unprotected DreamBooth, which successfully preserves the subject identity across diverse scenes. Although Anti-DreamBooth and DisDiff provide partial protection, the subject remains identifiable. In contrast, the proposed MAAT effectively disrupts identity features, producing unrecognizable outputs and significantly enhancing protection performance.

Wu, 2023). Second, current methodologies lack in-depth investigation into the timestep dynamics of diffusion models. Prevailing strategies either attack U-Net operations in single time steps (selected from full range (Van Le et al., 2023; Xu et al., 2024) or high gradient time steps (Wang et al., 2023; Liu et al., 2024)) or apply uniform temporal binning strategies that rigidly partition the diffusion process into fixed intervals (Tang et al., 2025). These approaches do not model the interdependencies and varying importance of timesteps during adversarial attacks, leading to suboptimal adversarial performance. Third, although the role of attention mechanisms has been acknowledged, existing approaches often overlook the layer-wise specialization involved in identity representation (Tang et al., 2025; Xu et al., 2024). Furthermore, the widely adopted ASPL framework (Van Le et al., 2023) suffers from distributional mismatch between clean-image pretrained surrogates and adversarial optimization phases, severely limiting attack transferability (Liu et al., 2023).

In this paper, we demonstrate that attacking diffusion models is equivalent to solving a multi-task optimization problem across timesteps. We show that sparse multi-task optimization in PGD exists: with proper clustering, representative tasks yield objectives within a controllable bound of the full problem, reducing conflicts, accelerating convergence, and approximating the Pareto front. Building on this insight, we propose an Adaptive Non-uniform Timestep Gradient Ensemble (ANTGE) to efficiently identify representative timesteps for constructing a sparse multi-task objective, significantly reducing computational overhead while preserving attack efficacy. Furthermore, we introduce a Layer-Aware Attention Targeting (LAT) Loss, which disrupts both self-attention and cross-attention mechanisms by selectively targeting the most identity-relevant attention layers. To accommodate the dynamic nature of model parameter updates during fine-tuning, we propose a two-stage Multi-timestep Alternating Adversarial Training (MAAT) paradigm. We evaluate MAAT in both white-box and black-box settings, achieving significant improvements in key metrics, including a 6.5% improvement in FDFR and an over 20% improvement in ISM. Runtime analysis shows that although MAAT takes slightly longer per iteration due to multiple gradients, it converges faster and achieves significantly better results with only a marginal increase in total time.

## 2 RELATED WORK

**Diffuion Models for Text-to-Image Generation.** Modern text-to-image diffusion models achieve state-of-the-art generation quality through architectural innovations and training strategies. Previous models like GLIDE (Nichol et al., 2021) introduced classifier-free guidance to enhance photorealism and text-image alignment. DALL·E 2 (Ramesh et al., 2022) further improved semantic coherence by integrating a diffusion prior with a CLIP-based text encoder. To balance image quality and computational efficiency, Imagen (Saharia et al., 2022) employ cascaded diffusion pipelines, while Latent Diffusion Models (LDMs) (Rombach et al., 2022) operate in compressed latent spaces using autoencoders. The integration of attention mechanisms, particularly cross-attention layers, facilitates the alignment between textual inputs and visual outputs. Stable Diffusion, a widely adopted open-source LDM, has democratized high-quality text-to-image synthesis across research and applications.

**Personalized Content Synthesis.** Personalized Content Synthesis (PCS) has emerged as a prominent aspect within the field of generative AI. In the era of diffusion model, Textual Inversion (Gal et al., 2022) was proposed to optimize a textual embedding of unique identifiers to represent the input concepts. DreamBooth (Ruiz et al., 2023), a popular diffusion-based personalization method, fine-tunes the pre-trained Stable Diffusion model using 3–5 reference images to link a rarely used identifier (e.g., "a sks person") to a new concept (e.g., a specific individual). Aiming to imporve finetuning efficiency, many studies focus on optimizing weight subsets or introducing additional adapters. For instance, CustomDiffusion (Kumari et al., 2023) only optimizes the cross-attention layers in the U-Net. SVDiff (Han et al., 2023) finetunes the singular values of weights. LoRa (Hu et al., 2021) accelerates the finetuning of personalized models by modifying cross-attention layers based on low-rank adaptation techniques. HyperDreamBooth (Ruiz et al., 2024) represents the input ID images as embeddings further improving the efficiency and speed of the personalization process. Recently, encoder-based methods that avoid fine-tuning have gained attention. Approaches like IP-Adapter (Ye et al., 2023), InstantID (Wang et al., 2024), and PhotoMaker (Li et al., 2024a) leverage general image encoders to extract ID features for conditioning the generation process.

**Privacy Protection for PCS.** To alleviate the privacy concerns caused by the malicious use of facial images through PCS methods based on stable diffusion, researchers have developed a series of proactive protection measures that embed adversarial noise in the user's images before release. Photoguard (Salman et al., 2023) maximizes the VAE-latent distance between protected and original images, preventing the diffusion model from faithfully reconstructing them. AdvDM (Liang et al., 2023) applies Projected Gradient Descent (PGD) (Madry et al., 2018) to attack the diffusion model's UNet model to prevent Textural Inversion. Anti-Dreambooth (Anti-DB) (Van Le et al., 2023) uses the Alternating Surrogate and Perturbation Learning (ASPL) to approximate the real trained models by alternately performing Dreambooth training and attack. SimAC (Wang et al., 2023) improves efficiency by greedily selecting timesteps with the highest gradient magnitude for perturbation updates. The CAAT (Xu et al., 2024) method proposes attacking only the K and V layers in the cross-attention structure of the diffusion model to improve the speed of generating adversarial samples. MetaCloak (Liu et al., 2023) tackles the bi-level poisoning issue in Anti-DB by employing a meta-learning framework and creates perturbations that are both transferable and robust. The DDAP (Yang et al., 2024) introduces a strategy that injects perturbations concurrently in both the spatial and frequency domains to enhance attack effectiveness. DisDiff (Liu et al., 2024) additionally set cross-attention erasure loss to erase the keyword's attention in attacking the Dreambooth process. PAP (Wan et al., 2024) proposes to model the distribution of prompts to generate universal adversarial perturbations, aiming to defend against diverse prompt-based attacks. DADiff (Tang et al., 2025) introduces a dual-stage attack that jointly optimizes prompt- and image-level perturbations, disrupting attention mechanisms and enhancing anti-customization via a local random-timestep gradient ensemble. Different from them, our MAAT introduces a novel two-stage training paradigm and formulates a sparse multi-task optimization framework to disrupt model learning, both grounded in the core objectives of diffusion models. We further consider both self-attention and cross-attention layers in diffusion models, and disrupt the alignment between text and image as well as the structural consistency of generated images by perturbing their attention maps.

## 3 METHODOLOGY

### 3.1 PRELIMINARY

**Text-to-Image Diffusion Models.** Diffusion models (Ho et al., 2020) consist of a noise-adding forward process and a denoising reverse process. During the forward process, noise is added at each timestep to generate a sequence $\{x_1, x_2, \ldots, x_T\}$, where $x_T$ is sampled from a standard normal distribution. A model $\epsilon_\theta(\cdot)$ is trained to predict the noise added to $x_t$, and is then used in the reverse process to infer $x_{t-1}$ from $x_t$, conditioned on the text prompt $y$. The training loss is $l_2$ distance:

$$L_{cond}(\theta, x_0) = E_{x_0, t, \epsilon \in \mathcal{N}(0,1)} \|\epsilon - \epsilon_\theta(x_{t+1}, t, y)\|_2^2, \quad (1)$$

where t is uniformly samples within $\{1, \ldots, T\}$. By iteratively sampling $x_{t-1}$, Gaussian noise $x_T$ is transformed into latent $x_0$. Latent Diffusion Models (LDMs) (Rombach et al., 2022), particularly the open-source Stable Diffusion variant, excel in text-to-image synthesis by operating in a compressed latent space. In our experiments, we leverage Stable Diffusion as the backbone model.

**Gradient-based adversarial attacks.** The adversarial attacks methods for diffusion models aim to prevent PCS technology from accurately learning image features during training. Existing methods (Liang et al., 2023; Van Le et al., 2023) attack the LDM (Rombach et al., 2022) by increasing the

loss during the training process, thus disrupting the fine-tuning process. The magnitude of these adversarial examples is typically constrained to be smaller than a constant value $\eta$. The determination of $\delta$ follows the formula below:

$$\delta = \arg \max_{\|\delta\|_p < \eta} L_{cond}(\theta, x + \delta). \tag{2}$$

The noise is limited to an $\eta - ball$ w.r.t. an $l_p$ metrics. Existing methods widely utilize Projected Gradient Descent (PGD) for iteratively optimizing adversarial examples. The process follows the formula below:

$$x^0 = x, \tag{3}$$

$$x^{n+1} = \Pi_{(x,\eta)}(x^n + \lambda \cdot \text{sgn}(\nabla_x L_{cond}(\theta, x + \delta, y))), \tag{4}$$

where $\lambda$ represents the step size during each iteration, and $n$ is the iteration number. With the operation $\Pi_{(x,\eta)}$, the noise is limited to an $\eta$-ball ensuring the adversarial examples are acceptable.

## 3.2 ADAPTIVE NON-UNIFORM TIMESTEP GRADIENT ENSEMBLE

In this subsection, we first demonstrate that adversarial attacks on diffusion models inherently constitute a multi-task optimization problem. Next, we theoretically and empirically verify the timestep transferability of adversarial perturbations. Finally, we introduce Adaptive Non-uniform Timestep Gradient Ensemble (ANTGE) to attack personalized generation with diffusion models.

### 3.2.1 DEFINITION OF THE MULTI-TASK OPTIMIZATION PROBLEM

**Lemma 1 (The equivalence of multi-task optimization)** *In adversarial attacks on diffusion models, maximizing the conditional loss function is equivalent to maximizing the sum of the losses over all timesteps:*

$$\max L_{cond}(\theta, x^{adv}) \equiv \max \sum_{t=1}^{T} L_{cond}(\theta, x_t^{adv}), \tag{5}$$

*where $x^{adv} = x + \delta$ and $L_t(\theta, x_t^{adv}) = E_{x_0^{adv}, \epsilon \in \mathcal{N}(0,1)} \|\epsilon - \epsilon_\theta(x_{t+1}^{adv}, t, y)\|_2^2$,*

According to Lemma1, we conceptualise each diffusion timestep $t$ as an independent sub-task endowed with its own loss function $L_{cond}(\theta, x_t)$. Consequently, adversarial attacks on diffusion models can be cast as a multi-task optimisation problem that seeks to maximise the aggregate loss across all sub-tasks. The derivation is provided in Appendix B.1.

However, diffusion models (Ho et al., 2020; Song et al., 2020; Rombach et al., 2022; Dhariwal & Nichol, 2021) are usually use $T = 10^3$ timesteps during training. Treating each timestep $t \in \{1, \ldots, T\}$ as an independent sub-task and computing adversarial losses across all $T$ timesteps per iteration incurs $\mathcal{O}(T)$ computational and memory overhead, which is impractical for real-time attacks. Moreover, as shown in Appendix D, gradients across timesteps often conflict, diminishing the effective averaged gradient and slowing convergence. This motivates the development of a principled approximation that retains the benefits of multi-task optimization while mitigating its computational and optimization drawbacks.

**Lemma 2 (Existence of Superior Sparse Multi-Task Optimization)** *Let $\{g_t(\delta)\}_{t=1}^{T}$ be the task gradients, where $g_t(\delta) = \nabla_x L_t(\theta, x_t^{adv})$, each smooth and bounded. Suppose there exists a partition $\{\mathcal{C}_k\}_{k=1}^{K}$ of $\{1, \ldots, T\}$ and representatives $r_k \in \mathcal{C}_k$ with $h_k(\delta) := g_{r_k}(\delta)$ such that*

$$\max_k \max_{t \in \mathcal{C}_k} \|g_t(\delta) - h_k(\delta)\| \leq \bar{\epsilon}. \tag{6}$$

*Then, under PGD updates with the representative gradients $\{h_k(\delta)\}_{k=1}^{K}$, the limit point $\delta^*$ satisfies*

$$\text{PD}(\delta^*) \lesssim \bar{\epsilon}, \tag{7}$$

*where $\text{PD}(\delta)$ denotes the Pareto-distance.*

*Moreover, compared to optimizing all $T$ tasks or performing sequential single-task updates, this sparse multi-task scheme reduces gradient conflicts and negative transfer, thereby converging to points that are closer to Pareto-stationary.*

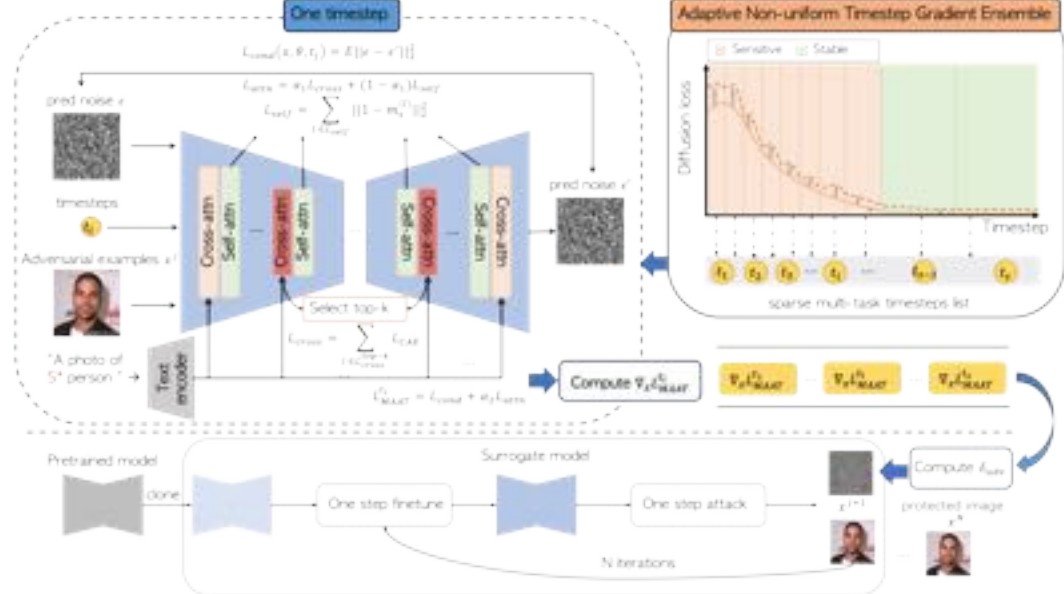

Figure 2: The pipeline of MAAT. During the training of the surrogate model, we only use adversarial examples. In the attack phase, we use the Adaptive Non-uniform Timestep Gradient Ensemble (ANTGE) to update adversarial noise. After N iterations of noise updates, we eventually obtain the protected images.

To address these challenges, we cluster the $T$ tasks and select representative timesteps to formulate a sparse multi-task optimization problem. Lemma 2 shows that the Pareto-distance of this sparse formulation admits a controllable upper bound relative to the Pareto-stationary solution of the full problem, determined by the within-cluster gradient discrepancy $\bar{\epsilon}$. Moreover, the sparse optimization scheme effectively reduces gradient conflicts, yielding more stable and consistent update directions across iterations. Therefore, it is crucial to design clustering rules that maximize within-cluster gradient consistency. The full derivation is provided in Appendix B.2.

### 3.2.2 DIFFUSION MODEL ATTACKING ACROSS TIMESTEPS

While Lemma B.2 establishes that grouping tasks with small gradient discrepancies ensures convergence to Pareto-stationary solutions, directly computing gradient similarities across all $T$ timesteps is computationally prohibitive. To bridge this gap between theory and practice, we systematically quantify each timestep's sensitivity to adversarial perturbations and its transferability across timesteps. Our key intuition is that timesteps exhibiting similar sensitivity–transferability patterns tend to share aligned gradient directions, thereby providing a tractable proxy for effective clustering.

**Sensitivity of timestep.** Prior studies Wang et al. (2023) reveal differences in adversarial gradient norms $\|\nabla_x L_{cond}(\theta, x_t, y)\|$ across diffusion timesteps: early steps (larger $t$) exhibit weaker gradients than final steps (smaller $t$), indicating heightened sensitivity to perturbations near generation completion.

**Transferability between different timesteps.** In the diffusion process, the adversarial perturbation $\delta_t$ generated for the sub-task at timestep $t$ propagates through the linear forward mapping and the weight-sharing denoiser, thereby exerting cross-task influence on other sub-tasks. Hence, the transferability of adversarial examples must be carefully considered. Because adjacent timesteps in the diffusion process have highly similar inputs, an adversarial perturbation $\delta$ retains part of its effectiveness when projected from $x_t$ to $x_{t-1}$, thereby enabling cross-step transferability. As the distance between two timesteps increases, input similarity diminishes and the corresponding transferability of the attack is expected to weaken.

**Definition 1** *Cross-step Attack Impact (CAI)* To quantify the generalization of an attack across different time steps, we define the Cross-step Attack Impact (CAI) for a pair of timesteps $(t, t') \in T^2$ as:

$$\text{CAI}(t, t') = L_{cond}(t', x_0 + \delta_t, \theta) - L_{cond}(t', x_0, \theta), \tag{8}$$

where $\delta_t$ is generated using only timestep $t$ during the attack, $t' \in [1, T]$.

The details of this evaluation and the results are presented in Appendix C. Experimental results reveal that a single attack $t$ yields consistently larger $\mathrm{CAI}(t, t')$ values for tasks at earlier timesteps. This arises can be attributed to two factors (i) smaller timestep $t'$ have intrinsically higher baseline losses, and (ii) they are more sensitive to perturbations, consistent with our timestep sensitivity analysis. As the attack timestep $t$ increases, the $\mathrm{CAI}(t, t')$ impact on earlier tasks gradually wanes, while its influence on later tasks correspondingly grows.

Analysis of the $\mathrm{CAI}(t, t')$ curves reveals two distinct phases. The first is the sensitive phase, corresponding to tasks with smaller attack timesteps. When a task's timestep is near the attack point, $\mathrm{CAI}(t, t')$ is high and decays steadily with increasing $|t - t'|$. This phase shows large peak-to-peak variations and pronounced divergence among the curves. The second is the stable phase corresponding to tasks with larger attack timesteps. In this stable phase, peak $\mathrm{CAI}(t, t')$ values differ only marginally, and the curves nearly overlap, indicating uniformly reduced sensitivity to the attack.

### 3.2.3 ANTGE: ADAPTIVE NON-UNIFORM TIMESTEP GRADIENT ENSEMBLE BY SPARSE MULTI-TASK OPTIMIZATION

Building upon the cross-timestep attack dynamics revealed earlier, we propose an Adaptive Non-uniform Timestep Gradient Ensemble (ANTGE) framework that fundamentally addresses two critical limitations in existing adversarial attacks: (i) neglect of sensitivity heterogeneity across timesteps, and (ii) insufficient consideration of cross-timestep adversarial transferability patterns. Therefore, we construct a sparse multi-task optimization framework by first performing $k$-shape clustering on the CAI trajectories, which yields $s$ clusters:

$$\left\{ (\mathcal{D}_s, \mu_s) \,\middle|\, \mathcal{D}_s = \left\{ \mathbf{c}_i \,\middle|\, \arg\max_s \left( \frac{\mathbf{c}_i - \bar{\mathbf{c}}_i}{\|\mathbf{c}_i - \bar{\mathbf{c}}_i\|} \cdot \mu_s \right)^2 = s \right\}, \, \|\mu_s\| = 1 \right\}_{s=1}^{S}. \tag{9}$$

where $\mathbf{c}_i \in \mathbb{R}^{1000}$ and $\mathbf{c}_i = [\mathrm{CAI}(t_i, 1), \mathrm{CAI}(t_i, 2), \ldots, \mathrm{CAI}(t_i, 1000)]^T$, $t_i \in \{10, 20, \ldots, 990\}$, $\bar{\mathbf{c}}_i$ is the mean value of the sequence $\mathbf{c}_i$, $\mathcal{D}_s$ is the $s$-th cluster containing CAI sequences with similar shapes, $\mu_s$ is centroid of the $s$-th cluster. The result shows that the sensitive phase forms many clusters, whereas the stable phase forms few, confirming our transferability and sensitivity analysis. During the optimization of the perturbation in the vicinity of the gradient $\delta$, we employ a multitask gradient aggregation rule:

$$\delta' \leftarrow \delta + \lambda \cdot \mathrm{sgn} \sum_{j=1}^{s} \nabla_x L_{\mathrm{cond}}(x_{t_j}, \theta, y). \tag{10}$$

where $\{t_j\}$ is sampled from each group $D_j(j = 1, \ldots, s)$. Aggregating cross-cluster gradients retains $O(s)$ complexity while capturing information from all timesteps, thus accelerating convergence and improving attack performance.

### 3.3 LAYER-AWARE ATTENTION TARGETING

Current anti-customization techniques typically perturb cross-attention layers uniformly, ignoring their distinct roles in encoding identity semantics (Liu et al., 2024; Tang et al., 2025). Therefore, we propose a Layer-Aware Attention Targeting (LAT) loss to disrupt critical attention layers during diffusion model fine-tuning. Our method separately addresses self-attention and cross-attention mechanisms with tailored objectives and layer selection strategies.

**Self-Attention Disruption Loss.** For self-attention maps $\mathrm{m}_s^{(l)}(x) \in \mathbb{R}^{H \times W \times C}$ at layer $l$, we maximize deviations in texture-critical layers $\mathcal{L}_{self}$ via:

$$\mathcal{L}_{\mathrm{self}} = \sum \left\| 1 - \mathrm{m}_s^{(l)}(x_{\mathrm{adv}}) \right\|_2^2 \tag{11}$$

This objective induces inconsistent feature aggregation, degrading fine-grained facial details.

**Cross-Attention Disruption Loss.** For identity-specific disruption, we first analyze the semantic relevance of cross-attention layers through mask-AUC correlation. During the model fine-tuning phase, we compute:

$$\mathrm{AUC}^{(l)} = \frac{1}{HW} \sum_{i,j} M_{i,j} \cdot \mathrm{m}_{i,j}^{(l)}(x, y) \tag{12}$$

where $M$ is the facial mask segmented by the SAM model (Kirillov et al., 2023) and $\mathrm{m}^{(l)}(x, y)$ represents the cross-attention map at layer $l$ for prompt $y$. Layers are ranked by their AUC scores, and

Figure 3: Quantitative results were compared with four classical methods, i.e., Anti-DB (Van Le et al., 2023), CAAT (Xu et al., 2024), SimAC (Wang et al., 2023) and DisDiff (Liu et al., 2024). The first row is "a photo of sks person", the second row is "a dslr portrait of sks person", the third row is "a photo of sks person in front of eiffel tower", the last row is "a photo of sks person looking at the mirror".

we select the top-$k$ layers as identity-critical layers. The adversarial loss for these layers is formulated as:

$$\mathcal{L}_{\text{cross}} = \sum_{l \in \mathcal{L}_{\text{top-k}}} L_{CAE}^l \tag{13}$$

where $L_{CAE}$ follows (Liu et al., 2024) to disrupt text-image alignmet. The total LAT loss combines both components:

$$\mathcal{L}_{\text{attn}} = \alpha_1 \cdot \mathcal{L}_{\text{self}} + (1 - \alpha_1) \cdot \mathcal{L}_{\text{cross}} \tag{14}$$

### 3.4 Multi-timestep Alternating Adversarial Training

Most state-of-the-art adversarial attacks adopt ASPL's three-stage pipeline (Van Le et al., 2023), but the clean-then-adversarial training sequence misaligns the surrogate with the target, greatly reducing the transferability of poisoning attacks (Liu et al., 2023). To address this issue, we introduce a two stage training procedure. As shown in Figure 2, in the first stage, the surrogate model $\theta'$ is pre-trained on adversarial images set $\mathcal{D}_{\text{adv}}$; in the second stage, $\theta'$ is used to update adversarial samples $x^{adv} \in \mathcal{D}_{\text{adv}}$. In sum, we propose the Multi-timestep Alternating Adversarial Training (MAAT), which centers on a co-evolution mechanism between the surrogate model and the adversarial samples. The adversarial attack process is shown below:

$$\theta' \leftarrow \theta.clone(), \tag{15}$$

$$\theta' \leftarrow \arg\min_{\theta'} \sum_{x \in \mathcal{D}_{adv}} L_{cond}(\theta', x), \tag{16}$$

$$L_{MAAT}(t_j) \leftarrow L_{\text{cond}}(t_j, x + \delta, \theta') + \alpha_2 L_{attn} \tag{17}$$

$$\delta_{adv} \leftarrow \delta + \lambda \cdot \text{sgn} \sum_{j=1}^{s} \nabla_x L_{MAAT}(t_j), \tag{18}$$

First, the backbone Stable Diffusion model $\theta$ (Eq. equation 15) is copied to $\theta'$. Then, the surrogate personalized generative model $\theta'$ is obtained by minimizing the adversarial loss $L_{cond}$. Subsequently, the adversarial perturbation $\delta$ is obtained by using ANTGE. This design eliminates the inductive bias introduced by initial clean pre-training, ensuring that the surrogate model is consistently exposed to adversarial distributions aligned with those encountered by the target model.

## 4 Experiments

### 4.1 Experimental Setup

**Datasets.** We use two facial datesets for experiments: VGG-Face2 (Cao et al., 2018) and CelebA-HQ (Karras et al., 2018) dataset. For CelebA-HQ and VGGFace2, we select subsets of 600 images, respectively, with each of 12 photos from an identical individual.

Table 1: Comparison with other open-sourced anti-personalization methods on datasets VGGFace2 (Cao et al., 2018) and CelebA-HQ (Karras et al., 2018).

| Dataset | Method | "a photo of sks person" | | | | "a dslr portrait of sks person" | | | |
|---|---|---|---|---|---|---|---|---|---|
| | | FDFR↑ | ISM↓ | BRISQUE↑ | FID↑ | FDFR↑ | ISM↓ | BRISQUE↑ | FID↑ |
| VGGFace2 | w/o Protect (Ruiz et al., 2023) | 0.06 | 0.56 | 15.61 | 236.37 | 0.21 | 0.48 | 7.61 | 279.05 |
| | Anti-DB (Van Le et al., 2023) | 0.35 | 0.47 | 39.53 | 402.12 | 0.36 | 0.34 | 33.68 | 448.98 |
| | SimAC (Wang et al., 2023) | 0.69 | 0.37 | 40.15 | 441.45 | 0.90 | 0.17 | 41.24 | 456.95 |
| | CAAT (Xu et al., 2024) | 0.69 | 0.27 | 41.92 | **474.76** | 0.76 | 0.20 | 39.56 | 459.86 |
| | DisDiff (Liu et al., 2024) | 0.68 | 0.29 | 39.72 | 410.16 | 0.94 | 0.10 | 41.07 | 467.94 |
| | **MAAT (Ours)** | **0.90** | **0.22** | **42.26** | 466.68 | **0.99** | **0.06** | **43.76** | **486.82** |
| CelebA-HQ | w/o Protect (Ruiz et al., 2023) | 0.07 | 0.63 | 13.15 | 154.63 | 0.30 | 0.46 | 8.75 | 221.89 |
| | Anti-DB (Van Le et al., 2023) | 0.35 | 0.47 | 39.53 | 386.06 | 0.36 | 0.34 | 33.68 | 384.84 |
| | SimAC (Wang et al., 2023) | 0.84 | 0.30 | 40.10 | 404.09 | 0.91 | 0.16 | 43.54 | 373.09 |
| | CAAT (Xu et al., 2024) | 0.31 | 0.47 | 42.46 | 315.63 | 0.36 | 0.36 | 27.26 | 311.57 |
| | DisDiff (Liu et al., 2024) | 0.80 | 0.33 | 38.61 | 437.19 | 0.94 | 0.11 | 42.40 | 480.38 |
| | **MAAT (Ours)** | **0.86** | **0.22** | **42.96** | **464.33** | **0.99** | **0.04** | **46.06** | **481.33** |

Table 2: Keyword mismatch which during training and testing on VGGFace2 dataset. The training prompt is "a photo of sks person" and the inference prompt is combined with the rare identifiers "sks" or "t@t".

| Dreambooth Prompt | Method | "a photo of S* person" | | | | "a dslr portrait of S* person" | | | |
|---|---|---|---|---|---|---|---|---|---|
| | | FDFR↑ | ISM↓ | BRISQUE↑ | FID↑ | FDFR↑ | ISM↓ | BRISQUE↑ | FID↑ |
| "sks"->"t@t" | Anti-DB (Van Le et al., 2023) | 0.36 | 0.40 | 41.88 | 344.12 | 0.45 | 0.31 | 35.13 | 347.44 |
| | SimAC (Wang et al., 2023) | 0.57 | 0.37 | 40.52 | 358.23 | 0.47 | 0.32 | 36.26 | 348.22 |
| | DisDiff (Liu et al., 2024) | 0.62 | 0.35 | 40.85 | 380.31 | 0.50 | 0.27 | 37.66 | 361.06 |
| | **MAAT (Ours)** | **0.63** | **0.29** | **42.97** | **393.39** | **0.56** | **0.25** | **42.23** | **377.73** |

**Implementation Details.** Our training regimen spans 30 epochs, each consisting of 3 steps to train the surrogate model and 6 steps to optimize the adversarial noise. The hyperparameters are set to $\alpha_1 = 0.5$ and $\alpha_2 = 0.1$, $\lambda_2 = 0.001$. In ANTGE, the diffusion timeline is partitioned into $s = 8$ clusters: the early stage ($t = 0$–240) is evenly split into 6 clusters, while the intermediate ($t = 240$–640) and late ($t = 640$–1000) stages each form a single cluster. This grouping is determined by the CAI-based clustering results, as described in Appendix G.5. We focus on the three attention layers ($k = 3$, {up_blocks.1.attentions.0, 1, 2}). The default configuration involves using Stable Diffusion v2.1 (Rombach et al., 2022) combined with DreamBooth (Ruiz et al., 2023). DreamBooth customizes models by reconstructing images using a generic prompt with pseudo-words "sks". For each prompt, we generated 16 images and used them to compute the metrics. Competing methods use their published default settings, and the noise budget is fixed at 0.05 for all approaches. Detailed hyper-parameters for the DreamBooth, LoRA (Hu et al., 2021), and TI (Gal et al., 2022) baselines are provided in the Appendix F.

**Evaluation Metrics.** We utilize 4 metrics, categorized into 2 aspects. (i) Similarity of the generated image to the input image: we use the Retinaface face detector (Deng et al., 2020) to evaluate the ratio of undetectable faces in the generated image as the **Face Detection Failure Rate (FDFR)**; if a face is detected, we use ArcFace (Deng et al., 2019) to encode it and calculate the average **Identity Score Matching (ISM)** with original images. (ii) Image quality: we utilize two widely used image quality metrics include **BRISQUE** (Mittal et al., 2012) and **FID** (Heusel et al., 2017).

## 4.2 COMPARISON WITH BASELINE METHODS

We conduct experiments on two datasets and compared them with existing open-source methods. We used two image generation prompts: one used during training, "a photo of sks people", and an unseen prompt, "a dslr portrait of sks people". As shown in Table 1, our proposed method outperforms other baseline methods, demonstrating superior comprehensive performance on two different datasets. On the white-box instance prompt ("a photo of a sks person"), compared to the baseline, our method MAAT yield better performance, with an increase of over 16% in the FDFR, and a decline of over 22% in the ISM. In the more challenging black-box instance prompt "a dslr portrait of sks person," MAAT pushes FDFR to a good result 0.99 where faces cannot be generated. Collectively, these findings indicate that MAAT achieves an optimal balance between privacy preservation. In terms of visualization, as shown in Figure 3, our method performs better in privacy protection compared to other baseline methods.

## 4.3 BLACK-BOX PERFORMANCE

We also investigate the performance of our method in black-box attacks, i.e., whether the proposed method is still effective when some components are unknown. All our experiments will use Stable Diffusion v2.1 and be conducted on the VGGFace2 dataset. We divide black-box attacks into three cases: 1) prompt mismatch, where the attacker uses different prompts; 2) model mismatch, where the attacker uses a different latent diffusion model; 3) personalization method mismatch, where

Table 3: Model versions mismatch during training and testing on VGGFace2 dataset. The training prompt is "a photo of sks person".

| Surrogate | Method | SD v1.5 | | | | SD v1.4 | | | |
|---|---|---|---|---|---|---|---|---|---|
| | | FDFR↑ | ISM↓ | BRISQUE↑ | FID↑ | FDFR↑ | ISM↓ | BRISQUE↑ | FID↑ |
| w/o Protect | (Ruiz et al., 2023) | 0.07 | 0.55 | 11.97 | 232.98 | 0.09 | 0.39 | 15.73 | 310.72 |
| v2.1 | Anti-DB (Van Le et al., 2023) | 0.82 | 0.10 | 45.96 | 349.72 | 0.86 | 0.09 | 38.42 | 368.05 |
| | SimAC (Wang et al., 2023) | 0.92 | 0.15 | 44.20 | 394.69 | 0.95 | 0.17 | 47.13 | 374.57 |
| | DisDiff (Liu et al., 2024) | 0.93 | 0.25 | 45.16 | 474.68 | 0.96 | 0.12 | 50.27 | 493.06 |
| | **MAAT (Ours)** | **0.99** | **0.10** | **54.97** | **531.80** | **0.99** | **0.05** | **60.03** | **518.52** |

Table 4: Customization method mismatch during training and testing on VGGFace2 dataset. The training is based on Dreambooth and the customization test is based on LoRA or TI.

| Train | Method | LoRA | | | | TI | | | |
|---|---|---|---|---|---|---|---|---|---|
| | | FDFR↑ | ISM↓ | BRISQUE↑ | FID↑ | FDFR↑ | ISM↓ | BRISQUE↑ | FID↑ |
| w/o Protect | | 0.06 | 0.58 | 11.90 | 284.05 | 0.06 | 0.38 | 2.73 | 222.70 |
| DreamBooth | Anti-DB (Van Le et al., 2023) | 0.64 | 0.23 | 42.07 | 403.38 | 0.43 | 0.12 | 36.79 | 289.66 |
| | DisDiff (Liu et al., 2024) | 0.95 | 0.04 | **39.69** | 491.83 | **0.83** | 0.06 | 52.84 | 487.74 |
| | **MAAT (Ours)** | **0.98** | **0.01** | 39.68 | **500.89** | **0.83** | 0.06 | **60.37** | **487.75** |

Table 5: The training time of the attackers under the default settings of MAAT and others.

| Method | Anti-DB | SimAC | Disdiff | MAAT (Ours) |
|---|---|---|---|---|
| time↓ | 126s | 226s | 198s | 223s |
| VRAM↓ | 18G | 20G | 20G | 20G |

the attacker uses a different personalization method. Additional robustness evaluations involving clean-perturbed image mixtures and image preprocessing are provided in the Appendix G.4.

**Prompt Mismatch.** When attackers employ stable diffusion for concept customization, the prompts they use during the addition of noise may vary from our initial assumptions. Therefore, we utilize the prompt "a photo of sks person" in our perturbation learning phase and replace the unique identifier "sks" with "t@t" during the fine-tuning of the DreamBooth model. The data shown in Table 2 reveal MAAT consistently outperforms prior methods across both prompt settings. Disrupting the self-attention module may reduce adversarial examples' reliance on specific prompt words, enhancing attack effectiveness under keyword mismatch.

**Model Mismatch.** We examine the effectiveness of adversarial noise learned on stable diffusion v2.1 against customization based on stable diffusion v1.4 and stable diffusion v1.5 in Table 3. The proposed MAAT method achieves state-of-the-art cross-model robustness, attaining the highest FDFR (0.99) and BRISQUE across SDS v1.5 and SD v1.4 frameworks, while maintaining competitive ISM. These results highlight our adaptive non-uniform timestep gradient ensemble method, which has strong model-agnostic adversarial efficacy.

**Customization Method Mismatch.** In the previous context, we assumed that malicious users fine-tuned the model based on DreamBooth. As shown in Table 4, we generate adversarial examples using the Dreambooth mechanism and then apply them for fine-tuning LoRA and Texture Inversion (TI). It indicates that, despite significant differences in fine-tuning principles, MAAT still achieves the best performance in adversarial attacks across mechanisms. This may be because the model structures employed by different fine-tuning mechanisms remain similar, and the destructive impact on the model's internal components is preserved across various generation mechanisms.

**Computational Overhead.** Table 5 reports computational overhead on a NVIDIA H100 (80 GB) with four same-ID images. MAAT finishes in 4 min—about 90 s slower than Anti-BD and <30 s slower than Disdiff. Although it computes eight gradients per step, yet its faster convergence keeps the extra overhead small and delivers superior results, showing that single-timestep attacks are sub-optimal. Further details of the computational complexity analysis are provided in Appendix H.

## 5 CONCLUSION

This work addresses the security risks associated with Personalized Content Synthesis in diffusion models. We propose a novel two-stage adversarial framework MAAT, which adaptively samples multiple non-uniform timesteps and jointly computes gradients to generate effective adversarial perturbations. In addition, we introduce a loss function that disrupts both self-attention and cross-attention mechanisms. Our contributions successfully render customized output unrecognizable and unidentifiable, outperforming state-of-the-art adversarial attack methods with a 6.5% improvement on FDFR and over 20% improvement on ISM.

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

## A  APPENDIX

## B  THEORETICAL DERIVATION

### B.1  THE EQUIVALENCE PROOF OF MULTI-TASK OPTIMIZATION

**Lemma 1**  In adversarial attacks on diffusion models, maximizing the conditional loss function is equivalent to maximizing the sum of the losses over all timesteps:

$$\max L_{cond}(\theta, x^{adv}) \equiv \max \sum_{t=1}^{T} L_t(\theta, x_t^{adv}) \tag{19}$$

where $x^{adv} = x + \delta$.

*Proof.* For each timestep $t \in [1, T]$, the loss function is defined as:

$$L_t(\theta, x_t^{adv}) = E_{x_0^{adv}, \epsilon \in \mathcal{N}(0,1)} \|\epsilon - \epsilon_\theta(x_{t+1}^{adv}, t, y)\|_2^2, \tag{20}$$

The conditional loss function of diffusion models is typically defined as the expectation of the training loss function:

$$L_{cond}(\theta, x^{adv}) \quad = E_{t \sim u(1,T)}[L_t(\theta, x_t^{adv})] \tag{21}$$

$$= \frac{1}{T} \sum_{t=1}^{T} L_t(\theta, x_t^{adv}) \tag{22}$$

Since $\frac{1}{T}$ is a constant factor and greater than zero, the optimization problem can be equivalently transformed into:

$$\max L_{cond}(\theta, x^{adv}, y) \equiv \max \sum_{t=1}^{T} L_t(\theta, x_t^{adv}, y) \tag{23}$$

## B.2 EXISTENCE OF SUPERIOR SPARSE MULTI-TASK OPTIMIZATION

**Lemma 2** Let $\{g_t(\delta)\}_{t=1}^{T}$ be the task gradients, where $g_t(\delta) = \nabla_x L_t(\theta, x_t^{adv})$, each smooth and bounded. Suppose there exists a partition $\{\mathcal{C}_k\}_{k=1}^{K}$ of $\{1, \ldots, T\}$ and representatives $r_k \in \mathcal{C}_k$ with $h_k(\delta) := g_{r_k}(\delta)$ such that

$$\max_k \ \max_{t \in \mathcal{C}_k} \|g_t(\delta) - h_k(\delta)\| \ \leq \ \bar{\epsilon}. \tag{24}$$

Then, under PGD updates with the representative gradients $\{h_k(\delta)\}_{k=1}^{K}$, any limit point $\delta^*$ satisfies
$$\mathrm{PD}(\delta^*) \lesssim \bar{\epsilon}, \tag{25}$$

where $\mathrm{PD}(\delta)$ denotes the Pareto-distance.

Moreover, compared to optimizing all $T$ tasks or performing sequential single-task updates, this sparse multi-task scheme reduces gradient conflicts and negative transfer.

*Proof.*

**Setup.** Let $F(\delta) := \frac{1}{T} \sum_{t=1}^{T} L_t(\theta, x_t^{adv})$ and $\bar{g}(\delta) := \nabla F(\delta) = \frac{1}{T} \sum_t g_t(\delta)$.

The correct PGD update with sign function is:

$$\delta^{(i+1)} = \Pi_{\|\delta\|_\infty \leq \eta} \left( \delta^{(i)} + \alpha \cdot \mathrm{sign}(\tilde{g}(\delta^{(i)})) \right), \tag{26}$$

where $\tilde{g}(\delta) = \frac{1}{T} \sum_{k=1}^{K} |\mathcal{C}_k| h_k(\delta)$ is our representative-based gradient approximation.

The approximation bias satisfies:

$$\|b(\delta)\| = \|\tilde{g}(\delta) - \bar{g}(\delta)\| = \left\| \frac{1}{T} \sum_{k=1}^{K} \sum_{t \in \mathcal{C}_k} (h_k(\delta) - g_t(\delta)) \right\| \leq \bar{\epsilon}. \tag{27}$$

**Pareto-distance Definition.** The Pareto-distance at $\delta$ is defined as:

$$\mathrm{PD}(\delta) = \min_{\lambda \in \Delta^{T-1}} \left\| \sum_{t=1}^{T} \lambda_t g_t(\delta) \right\|, \tag{28}$$

where $\Delta^{T-1}$ is the probability simplex. Since the uniform weight vector $\lambda_t = 1/T$ is a feasible point, we have:

$$\mathrm{PD}(\delta) \leq \left\| \sum_{t=1}^{T} \frac{1}{T} g_t(\delta) \right\| = \|\bar{g}(\delta)\|. \tag{29}$$

**One-step Analysis with Sign Function.** For PGD with sign function, we analyze the inner product between the true gradient and the update direction:

Let $u(\delta) = \mathrm{sign}(\tilde{g}(\delta))$ be the update direction. We have:
$$\langle \bar{g}(\delta), u(\delta) \rangle = \langle \bar{g}(\delta), \mathrm{sign}(\tilde{g}(\delta)) \rangle. \tag{30}$$

Since $\tilde{g}(\delta) = \bar{g}(\delta) + b(\delta)$ with $\|b(\delta)\| \leq \bar{\epsilon}$, we can bound this inner product:

$$\langle \bar{g}(\delta), \text{sign}(\tilde{g}(\delta)) \rangle = \frac{\langle \bar{g}(\delta), \tilde{g}(\delta) \rangle}{\|\tilde{g}(\delta)\|} \quad \text{(by properties of sign function in PGD)} \tag{31}$$

Expanding the numerator:
$$\langle \bar{g}(\delta), \tilde{g}(\delta) \rangle = \|\bar{g}(\delta)\|^2 + \langle \bar{g}(\delta), b(\delta) \rangle \geq \|\bar{g}(\delta)\|^2 - \|\bar{g}(\delta)\|\|b(\delta)\| \geq \|\bar{g}(\delta)\|^2 - \bar{\epsilon}\|\bar{g}(\delta)\|. \tag{32}$$

For the denominator, using reverse triangle inequality:
$$\|\tilde{g}(\delta)\| \geq \|\bar{g}(\delta)\| - \|b(\delta)\| \geq \|\bar{g}(\delta)\| - \bar{\epsilon}. \tag{33}$$

Thus, when $\|\bar{g}(\delta)\| > \bar{\epsilon}$, we have:
$$\langle \bar{g}(\delta), u(\delta) \rangle \geq \frac{\|\bar{g}(\delta)\|^2 - \bar{\epsilon}\|\bar{g}(\delta)\|}{\|\bar{g}(\delta)\| - \bar{\epsilon}} = \|\bar{g}(\delta)\|. \tag{34}$$

**Lipschitz Constant and Step Size.** The Lipschitz constant $L$ comes from the smoothness assumption: each $L_t$ is $L$-smooth, meaning:
$$\|g_t(\delta) - g_t(\delta')\| \leq L\|\delta - \delta'\| \quad \text{for all } \delta, \delta'. \tag{35}$$

With step size $\alpha = 0.001$ (a typical small constant for PGD), and using the descent lemma for non-smooth analysis:

The one-step progress can be bounded as:
$$F(\delta^{(i+1)}) - F(\delta^{(i)}) \geq \alpha\langle \bar{g}(\delta^{(i)}), u(\delta^{(i)}) \rangle - \frac{L\alpha^2}{2}. \tag{36}$$

Substituting our lower bound:
$$F(\delta^{(i+1)}) - F(\delta^{(i)}) \geq \alpha\|\bar{g}(\delta^{(i)})\| - \frac{L\alpha^2}{2}, \tag{37}$$

when $\|\bar{g}(\delta^{(i)})\| > \bar{\epsilon}$.

**Limit-point Analysis.** Since the sequence $\{\delta^{(i)}\}$ is bounded (projected to $\|\delta\|_\infty \leq \eta$) and $F$ is continuous, there exists a convergent subsequence $\delta^{(i_j)} \to \delta^*$.

At the limit point $\delta^*$, the one-step progress must vanish:
$$\lim_{j \to \infty} \left[ F(\delta^{(i_j+1)}) - F(\delta^{(i_j)}) \right] = 0. \tag{38}$$

This implies:
$$\alpha\|\bar{g}(\delta^*)\| - \frac{L\alpha^2}{2} \leq 0 \quad \Rightarrow \quad \|\bar{g}(\delta^*)\| \leq \frac{L\alpha}{2}. \tag{39}$$

If $\|\bar{g}(\delta^*)\| \leq \bar{\epsilon}$, then the bound is immediate. Otherwise, from our earlier analysis:
$$\|\bar{g}(\delta^*)\| \leq \frac{L\alpha}{2}. \tag{40}$$

**Pareto-distance Bound.** Since $\text{PD}(\delta) \leq \|\bar{g}(\delta)\|$ for the uniform weight vector, we have:
$$\text{PD}(\delta^*) \leq \|\bar{g}(\delta^*)\| \leq \max\left( \bar{\epsilon}, \frac{L\alpha}{2} \right). \tag{41}$$

Given that $\alpha = 0.001$ is typically chosen to be small relative to the problem scale, and $\bar{\epsilon}$ represents the clustering approximation error, we obtain:
$$\text{PD}(\delta^*) \lesssim \bar{\epsilon}. \tag{42}$$

**Conflict Reduction Analysis.** The key advantage of our clustering approach is that within each cluster, gradients are aligned (up to $\bar{\epsilon}$ error). The sign function in PGD amplifies this alignment:

- When gradients within a cluster point in similar directions, their signs agree - When using all $T$ gradients, opposing gradients may cancel in the sign operation - Our approach preserves the dominant direction within each cluster

**Remark on conflict reduction.** Let

$$C(\delta) := \frac{1}{T(T-1)} \sum_{t \neq t'} \frac{\langle g_t(\delta), g_{t'}(\delta) \rangle}{\|g_t(\delta)\|\|g_{t'}(\delta)\|} \tag{43}$$

denote the average cosine similarity between task gradients. A lower value of $C(\delta)$ indicates stronger gradient conflicts.

The clustering strategy alleviates this issue by ensuring that within each cluster, gradients remain aligned up to an error of $\bar{\epsilon}$. Consequently, when representative gradients are used in PGD:

- Within-cluster gradients share similar directions, and their sign patterns reinforce each other.

- In contrast, when all $T$ gradients are aggregated, opposing directions may cancel out during the sign operation.

- By focusing on representatives, the updates preserve the dominant orientation of each cluster and suppress conflicting signals.

This mechanism effectively reduces gradient conflict, yielding more stable and consistent update directions across iterations.

## C    ANALYSIS OF DIFFUSION MODEL ATTACKING ACROSS TIMESTEPS

To examine the cross-temporal generalization capability of adversarial attacks, we conducted a comprehensive analysis using the CAI metric.

**Experimental Setup.**

(1) Dataset and Model Configuration. We conduct experiments on the high-resolution CelebA-HQ dataset (512×512). A total of 500 images are randomly selected and preprocessed via standard normalization. The base model used is Stable Diffusion v2.1-base. To reduce computational costs, we adopt a sparse timestep sampling strategy with $T_1 = \{t \mid t = 10i, \ i \in \mathbb{N}, \ 10 \leq t \leq 990\}$ and $T_2 = [1, 999]$, excluding the outermost 2% of timesteps to avoid boundary effects.

(2) Attack Algorithm. We follow the Anti-DreamBooth framework, where adversarial examples are iteratively generated by alternating between surrogate model finetuning and perturbation optimization. The experimental procedure is summarized in Algorithm 1.

**Result.** A subset of the results is shown in Fig 4, where the horizontal axis represents $t'$ and the vertical axis corresponds to $CAI(t, t')$. As show in Figure 4, a single adversarial perturbation induces significantly higher CAI values for early-stage denoising tasks compared to later stages. This phenomenon is attributed to two intrinsic mechanisms: (1) the inherently elevated baseline losses in early timesteps, which dominate low-frequency structural reconstruction, and (2) the exponential decay of temporal sensitivity, quantified by gradient magnitude analysis. Furthermore, a bidirectional transfer pattern emerges: as the attack shifts to later timesteps, the CAI influence on early tasks diminishes while its effect on subsequent stages increases proportionally. This behavior aligns with the Lipschitz-constrained evolution of the potential space in diffusion models, highlighting the critical role of early timesteps in adversarial vulnerability.

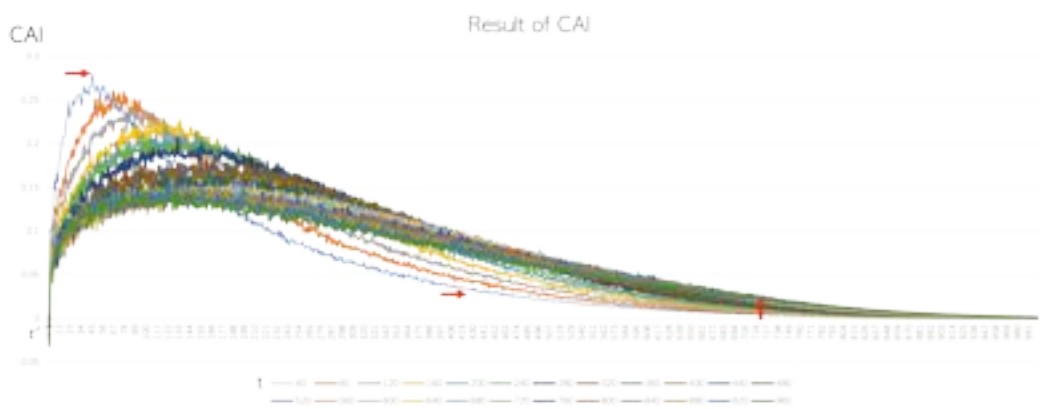

Figure 4: Result of Cross-Attention Interference (CAI).

---

**Algorithm 1** Computation of Cross-Attention Interference (CAI)

1: **Input:** Stable Diffusion model $\theta$, clean image $X_{\text{cl}}$, prompt $p$, surrogate finetune steps $s_1$, PGD steps $s_2$, step size $\lambda$
2: **Output:** Protected image $X_{\text{adv}}$
3: **function** SURROGATEMODEL($\theta, X, p$)
4:     Freeze all layers except the text encoder and cross-attention modules in the U-Net
5:     Randomly sample a timestep $t$
6:     $\theta' \leftarrow \arg\min_{\theta'} \mathcal{L}_{DM}(X, p, t)$
7:     **return** $\theta'$
8: **end function**
9: **for** $s = 1$ to $s_1$ **do**
10:     $\theta' \leftarrow$ SURROGATEMODEL($\theta, X_{\text{cl}}, p$)
11: **end for**
12: $X_{\text{adv}} \leftarrow X_{\text{cl}}$
13: **for** $s = 1$ to $s_2$ **do**
14:     **for** $t \in T_1$ **do**
15:         **for** $t' \in T_2$ **do**
16:             Compute CAI($t, t'$)
17:         **end for**
18:         Randomly sample $t \in T$
19:         $\delta_{\text{adv}} \leftarrow \arg\max_{\delta} L_{\text{cond}}(f_\theta(X_{\text{adv}}, p))$
20:         $X_{\text{adv}} \leftarrow X_{\text{adv}} + \delta_{\text{adv}}$
21:     **end for**
22: **end for**

---

## D EXPERIMENTAL VERIFICATION OF GRADIENTS CONFLICT ACROSS TIMESTEPS

To validate the existence of the gradients conflict, we formally define the Gradient Alignment Score (GAS) metric, which quantifies the cosine similarity between gradients across diffusion timesteps:

**Definition 2** *Gradient Alignment Score (GAS)*. To quantify the consistency of optimization directions across timesteps, we define the *Gradient Alignment Score (GAS)* for a pair of timesteps $(t, t') \in \mathcal{T}^2$ as:

$$\text{GAS}(t, t') = \frac{\langle \nabla_x \mathcal{L}_{\text{cond}}(x, \theta, t),\ \nabla_x \mathcal{L}_{\text{cond}}(x, \theta, t') \rangle}{\|\nabla_x \mathcal{L}_{\text{cond}}(x, \theta, t)\| \cdot \|\nabla_x \mathcal{L}_{\text{cond}}(x, \theta, t')\|}. \tag{44}$$

This metric reflects the directional consistency of gradients in the latent space across different diffusion timesteps. The computation of Gradient Alignment Score (AGS) follows a similar procedure

to that of Cross-Attention Interference (CAI) as Algorithm 1, with the key difference being that $T_1 = T_2 = \{t \mid t = 10i, \ i \in \mathbb{N}, \ 10 \leq t \leq 990\}$ .

**Results and Observations.** As shown in Figure 5a, to quantify the alignment properties of adversarial gradients across diffusion timesteps, we analyze the distribution of the GAS and derive three key insights: (1) Coverage: 85(2) Directional Consistency: Within the covered subset $\mathcal{T}_+$, the average cosine similarity of gradient directions across timesteps reaches $\gamma_2 = 0.40$, significantly exceeding the random baseline ($\gamma_{random} \approx 0$), validating the statistical alignment advantage; (3) Theory-Experiment Consistency: Empirical results support the theoretical lower bound of loss increment:

$$\mathbb{E}_{t,t'} \left[ \Delta L_{\text{cond}}(\theta, x_{t'}, y) \right] \geq lambda \gamma_1 \lambda_2 \cdot \|\nabla_x \mathcal{L}\|_{\text{avg}}. \tag{45}$$

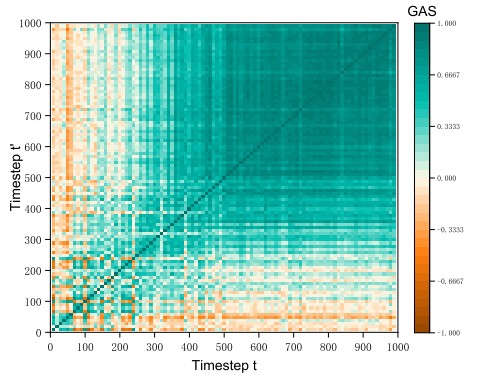

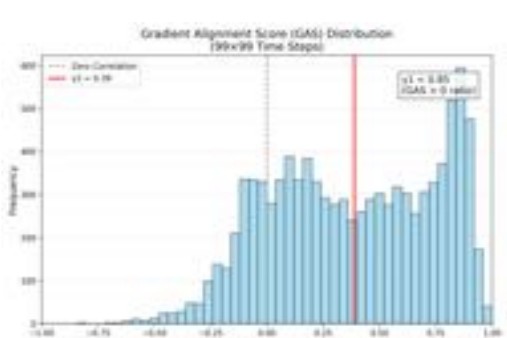

(a) Heatmap of GAS Across Diffusion Timesteps.

(b) Histogram and Statistical Distribution of GAS Across Diffusion Timesteps.

Figure 5: Visualization and Statistical Characterization of GAS on CelebA-HQ Dataset.

# E  LAYER-WISE AUC ANALYSIS OF CROSS-ATTENTION MAPS FOR IDENTITY-TARGETED ATTACKS

During the fine-tuning stage of diffusion models, we analyzed the cross-attention maps within the U-Net architecture that are conditioned on the instant prompt. Specifically, we leveraged segmentation masks generated by the Segment Anything Model (SAM) to annotate target facial regions and computed the Area Under the Curve (AUC) scores between the attention maps and the corresponding facial masks. As shown in Figure 6, we selected the top-3 attention layers with the highest AUC scores as our attack targets, as they demonstrated the strongest alignment with identity-relevant regions compared to other layers.

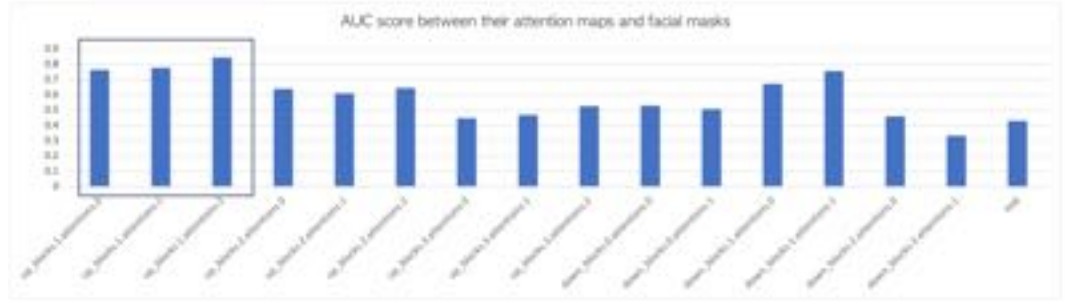

Figure 6: AUC-Based Analysis of Cross-Attention Maps and Facial Masks in Stable Diffusion.

Table 6: Hyperparameters of diffusion models, which follow their default configurations. DB and TI denote DreamBooth, and Textual Inversion, respectively.

| parameters | DB | TI | LoRA |
|---|---|---|---|
| train steps | 1000 | 3000 | 500 |
| learning rate | $5 \times 10^{-7}$ | $5 \times 10^{-4}$ | $1 \times 10^{-4}$ |
| batchsize | 2 | 1 | 1 |

## F  IMPLEMENTATION DETAILS

**Models.** Since the most popular open-source LDM is Stable Diffusion, our experiments are mainly conducted on the latest version of Stable Diffusion 2.1. To test the performance of our method in a black-box scenario, we set two conditions: one assumes that the model version used by the user is the same as the version we used in the experiment, and the other assumes that the version used by the user is different from ours. We conducted corresponding experiments in both scenarios.

**Baselines.** The baseline methods we selected (including Anti-DB, SimAC, CAAT and Disdiff) are the current state-of-the-art methods in the field of addressing the misuse of PCS in text-to-image generation. These methods have been published at top conferences in the relevant field, demonstrated strong performance on multiple benchmark datasets, and are widely used in research on combating PCS. For fair comparison, we reproduced their source codes and trained adversarial samples under the same noise budget and focused on comparing their performance in untargeted scenarios. This approach avoids the bias introduced by selecting specific target images, as each method may have its own optimized target image. These protected samples were then used to fine-tune the diffusion model (DreamBooth) with the prompt "a photo of sks person" . The fine-tuned diffusion model was subsequently tested using the prompts "a photo of sks person" and "a dslr portrait of sks person."

**Hyperparameters.** Detailed hyper-parameters for the DreamBooth, LoRA, and Textual Inversion (TI) baselines are provided in Table 6. For DreamBooth, the text encoder and UNet model were trained with a batch size of 2 and a learning rate of $5 \times 10^{-7}$ for 1,000 steps. Textual Inversion is trained with a batch size of 1 and a learning rate of $5 \times 10^{-4}$ for 3,000 steps. LoRA fine-tunes the text encoder and U-Net with a batch size of 1 and a learning rate of $1 \times 10^{-4}$ for 500 steps.

## G  ADDITIONAL RESULTS

### G.1  QUANTITATIVE EVALUATION OF IMAGE QUALITY AFTER PERTURBATION

We evaluated the image quality of clean images after adding adversarial perturbations, with detailed results presented in Table 7. For our method, the LPIPS is 0.16 and the PSNR is 36.68dB (with differences within 0.1 and 0.5, respectively, compared to the best baselines). An LPIPS value of 0.16 indicates that the perceptual distance between adversarial samples and the original images in the deep feature space is small, which is considered high quality in mainstream generative and adversarial attack literature. Moreover, a PSNR of 36dB is well above the commonly accepted threshold of 30dB for visually lossless quality, further confirming that the perturbations have a very limited impact on the overall structure and details of the images.

Table 7: Quantitative evaluation of visual quality degradation after adversarial perturbation.

| Method | CelebA-HQ | | VGGFace2 | |
|---|---|---|---|---|
| | LPIPS↓ | PSNR↑ | LPIPS↓ | PSNR↑ |
| Anti-DB | 0.21 | 36.92 | 0.18 | 36.53 |
| SimAC | **0.15** | 37.10 | **0.17** | 37.14 |
| CAAT | 0.37 | 32.67 | 0.42 | 32.52 |
| Disdiff | **0.15** | **37.11** | **0.17** | **37.17** |
| MAAT | 0.16 | 36.68 | 0.18 | 36.76 |

### G.2 ASSESSMENT OF REMAINING FACIAL FEATURES AFTER INTERFERENCE

To assess how many facial features remain after perturbation, we use the SER-FIQ metric, which quantitatively evaluates the quality and robustness of facial representations under distortion. SER-FIQ is widely adopted for this purpose and is well suited to our scenario. Lower SER-FIQ scores indicate less distinguishable facial features, effectively reflecting the extent of feature removal. As shown in Table 8, our method achieves significantly lower SER-FIQ scores than other approaches.

Table 8: SER-FIQ ($\downarrow$) scores comparison for different methods on VGGFace2 dataset.

| Prompt | Anti-DB | SimAC | Disdiff | MAAT |
|---|---|---|---|---|
| "a photo of sks person" | 0.43 | 0.21 | 0.29 | **0.19** |
| "a dslr portrait of sks person" | 0.47 | 0.25 | 0.18 | **0.09** |

### G.3 ABLATION STUDY

**Sparse Timestep Ablation** We proposed Adaptive Non-uniform Timestep Gradient Ensemble (ANTGE) framework, grounded in sparse multi-task optimization, systematically addresses the critical trade-off between computational efficiency and adversarial efficacy. To determine the optimal number of subtasks required to approximate full-task performance, we conducted ablation studies across varying task subset sizes and further compared ANTGE with a Local Random Timestep Gradient Ensemble (LRTGE) baseline. Unlike ANTGE's adaptive timestep selection, LRTGE uniformly partitions all timesteps into 25 groups and randomly samples tasks from these groups to construct a 25-task optimization problem. Full-Attack refers to jointly optimizing all $T$ tasks simultaneously. For the ANTGE/14*, we split the sensitive region into 12 clusters and preserved the remaining 2 clusters as in ANTGE/8, resulting in 14 total clusters. Experimental results in Table 9 demonstrate that ANTGE achieves superior performance with only 8 subtasks, outperforming LRTGE in attack success rate (FDFR: 0.93 vs. LRTGE's 0.91) while reducing gradient computation overhead by 67%. This validates ANTGE prioritizes timesteps with high sensitivity to adversarial perturbations, avoiding the redundancy inherent in uniform sampling. Our non-uniform weighting strategy amplifies gradients from critical timesteps, whereas LRTGE's uniform aggregation dilutes attack potency.

Table 9: Sparse Timestep Ablation.

| Method | Average | | | |
|---|---|---|---|---|
| | FDFR$\uparrow$ | ISM$\downarrow$ | BRISQUE$\uparrow$ | FID$\uparrow$ |
| Full-Attack | 0.84 | 0.28 | 38.67 | 449.75 |
| LRTGE/25 | 0.91 | 0.16 | 42.75 | 464.52 |
| ANTGE/14* | **0.94** | **0.14** | 42.74 | 472.60 |
| ANTGE/10 | 0.91 | 0.21 | 43.49 | 465.09 |
| ANTGE/8 | **0.94** | **0.14** | **43.76** | **474.79** |
| ANTGE/6 | 0.87 | 0.25 | 40.90 | 420.12 |

To validate the effectiveness of clustering based on the CAI metric, we further analyzed the relationship between CAI and attack performance. Specifically, we computed the total post-attack CAI values under different attack settings:

$$\sum CAI = \sum_{t'=1}^{999} \big[ L_{\text{cond}}(t', x + \delta, \theta) - L_{\text{cond}}(t', x, \theta) \big] \tag{46}$$

The results in Table 10 reveal a clear positive correlation between the total CAI value and attack performance, confirming that CAI serves as an effective clustering criterion for improving empirical results.

Although an ideal multi-task optimization where full-attack is optimal, real-world attacks must contend with: (1) heterogeneous gradients across timesteps, (2) a limited perturbation budget $\delta$, and

(3) diminishing returns on low-sensitivity steps. Our empirically derived CAI metric overcomes these issues by concentrating the budget on the most impactful timesteps.

The performance of the ANTGE/14* configuration is nearly indistinguishable from that of ANTGE/8, suggesting that ANTGE/8 already covers virtually all of the most sensitive timesteps. This implies that once the number of clusters in the sensitive region surpasses a certain threshold, adding more clusters produces only negligible changes in CAI—because the timesteps they contain exhibit almost identical sensitivities to those in the existing clusters—so further increasing the cluster count yields no additional improvements.

Table 10: Sum of CAI values under different attack schemes.

|  | LRTGE/25 | Full-Attack | ANTGE/6 | ANTGE/8 | ANTGE/10 | ANTGE/14* |
|---|---|---|---|---|---|---|
| $\sum$ CAI | 107.25 | 110.31 | 78.05 | 125.47 | 89.75 | 127.29 |

**Evaluations between Different Losses** The ablation study in Table 11 demonstrates significant differences in the effectiveness of adversarial attacks under varying loss combinations. When introducing the self-attention loss reduces ISM to 0.31, yet the FDFR drops from 0.69 to 0.64, suggesting that optimizing self-attention alone may weaken interference with other features. In contrast, the combination of cross-attention loss with $L_{cond}$ significantly improves FDFR and BRISQUE, validating the critical role of cross-attention layers in disrupting identity semantics. The proposed MAAT method, which jointly optimizes cross- and self-attention losses alongside dynamic training strategies, achieves the best performance across all metrics. These results confirm that the synergistic design of layer-aware attention targeting loss optimization comprehensively disrupts identity generation pathways while enhancing adversarial robustness.

Table 11: Ablation study of using different losses.

| Method | Average | | | |
|---|---|---|---|---|
|  | FDFR↑ | ISM↓ | BRISQUE↑ | FID↑ |
| $L_{\mathrm{cond}}$ | 0.69 | 0.37 | 39.21 | 442.10 |
| $L_{\mathrm{cond}} + L_{self}$ | 0.64 | 0.31 | 34.86 | 431.48 |
| $L_{\mathrm{cond}} + L_{cross}$ | 0.85 | 0.28 | 40.90 | 451.65 |
| **MAAT** | **0.94** | **0.14** | **43.76** | **474.79** |

**Comparison with ASPL Training Strategy** We further performed additional ablation experiments on the VGGFace2 dataset to directly compare MAAT and ASPL. As presented in Table 12, the "ASPL baseline" refers to our MAAT method trained using the three-stage pipeline as described in the original ASPL framework, while the "MAAT" row uses our proposed two-stage training procedure. All other components (including attack algorithm, loss) remain consistent across both settings. The comparison in Table 12 thus isolates the effect of the training strategy alone. Our results demonstrate that the two-stage MAAT pipeline brings consistent improvements over the three-stage (ASPL-style) pipeline on the VGGFace2 dataset, confirming the effectiveness of our proposed approach.

Table 12: Comparison between MAAT and ASPL on VGGFace2 dataset.

| Method | Prompt | FDFR↑ | ISM↓ | BRISQUE↑ | FID↑ |
|---|---|---|---|---|---|
| ASPL | a photo of sks person | 0.85 | 0.27 | 41.42 | 458.25 |
| MAAT | a photo of sks person | **0.90** | **0.22** | **42.26** | **466.68** |
| ASPL | a dslr portrait of sks person | 0.93 | 0.12 | 42.69 | 482.35 |
| MAAT | a dslr portrait of sks person | **0.99** | **0.06** | **43.79** | **486.82** |

## G.4 ROBUST ANALYSIS

**Robustness Evaluation under Clean-Perturbed Image Mixtures.** Our experiments thus far have been conducted under the idealized assumption that all subject images used for training are embedded

with protective perturbations. In this section, we relax this assumption to investigate a more realistic scenario where clean and perturbed images are mixed during DreamBooth fine-tuning. Specifically, we fix the total number of training images per identity to four and examine three configurations in which the number of clean images increases from one to three (see Table 13). Experimental results show that our defense remains effective when at least half of the training images are perturbed. However, its effectiveness gradually declines as the proportion of clean images increases.

Table 13: Defense performance of MAAT on VGGFace2 in uncontrolled settings. We include two extra results with 0 clean image (convenient setting) and 0 perturbed image (no defense) for comparison.

| Perturbed | Clean | "a photo of *sks* person" | | | | "a dslr portrait of *sks* person" | | | |
|---|---|---|---|---|---|---|---|---|---|
| | | FDFR↑ | ISM↓ | BRISQUE↑ | FID↑ | FDFR↑ | ISM↓ | BRISQUE↑ | FID↑ |
| 4 | 0 | **0.90** | **0.22** | **42.26** | **466.68** | **0.99** | **0.06** | **43.76** | **486.82** |
| 3 | 1 | 0.50 | 0.43 | 35.53 | 332.05 | 0.52 | 0.35 | 34.01 | 406.85 |
| 2 | 2 | 0.29 | 0.53 | 28.99 | 287.80 | 0.40 | 0.37 | 26.13 | 334.20 |
| 1 | 3 | 0.11 | 0.61 | 17.77 | 245.93 | 0.26 | 0.43 | 12.66 | 283.89 |
| 0 | 4 | 0.06 | 0.56 | 15.61 | 236.37 | 0.21 | 0.48 | 7.61 | 279.05 |

**Attacks after image pre-processings.** Before being released online, images are typically preprocessed to reduce storage costs, with JPEG compression and Gaussian blur being two widely adopted techniques. To evaluate the robustness of our defense under such common image preprocessing techniques, we apply both to the protected images. As shown in Table 14, these transformations indeed weaken the effectiveness of the defense to some extent. Nevertheless, the quality degradation of the generated images remains substantial, as indicated by consistently high BRISQUE scores across different settings. These results suggest that our defense maintains a reasonable level of robustness against typical image preprocessing operations.

Table 14: MAAT's performance on VGGFace2

| | "a photo of sks person" | | | | "a dslr portrait of sks person" | | | |
|---|---|---|---|---|---|---|---|---|
| | FDFR↑ | ISM↓ | BRISQUE↑ | FID↑ | FDFR↑ | ISM↓ | BRISQUE↑ | FID↑ |
| MAAT | 0.85 | 0.22 | 42.52 | 454.00 | 0.98 | 0.05 | 44.54 | 480.72 |
| Gaussian Blur K=3 | 0.43 | 0.41 | 46.44 | 335.25 | 0.64 | 0.26 | 43.71 | 396.47 |
| Gaussian Blur K=5 | 0.13 | 0.53 | 47.67 | 251.26 | 0.26 | 0.37 | 36.64 | 297.55 |
| Gaussian Blur K=7 | 0.07 | 0.55 | 47.93 | 229.47 | 0.22 | 0.43 | 24.28 | 239.85 |
| JPEG Comp. Q=30 | 0.06 | 0.55 | 34.58 | 193.29 | 0.22 | 0.44 | 9.74 | 234.40 |
| JPEG Comp. Q=50 | 0.07 | 0.51 | 33.01 | 218.58 | 0.22 | 0.42 | 12.44 | 240.25 |
| JPEG Comp. Q=70 | 0.13 | 0.45 | 39.74 | 261.13 | 0.23 | 0.42 | 23.72 | 288.58 |
| No def., no preproc. | 0.06 | 0.56 | 15.61 | 236.37 | 0.21 | 0.48 | 7.61 | 279.05 |

**Results on More Inference Prompts.** Due to space constraints, we only present the quantitative results on the white-box instance prompt "a photo of sks person" and a black-box inference prompt "a dslr portrait of sks person" in Table 1 of the main text. To further demonstrate the effectiveness of MAAT in cross-prompt transferability, we generate adversarial examples using the white-box instance prompt "a photo of sks person", and present quantitative comparisons in Table 15 for the other three black-box inference prompts: "a photo of sks person looking at the mirror", "a photo of sks person in front of Eiffel Tower", and "a photo of sks person sitting on the chair". MAAT achieves the best results in all metrics on all blackbox inference prompts.

However, in real-world scenarios, prompts are often more complex and diverse. To better reflect such conditions, we further conducted experiments on the VGGFace2 dataset using three semantically richer and more diverse prompts:

Prompt 1 – "Candid shot of sks person just chillin' like a real person: leaning against a cafe window on a rainy afternoon, absently stirring a coffee while lost in thought, with soft reflections of neon lights on the glass."

Table 15: Performance comparison on VGGFace2 dataset with diverse prompts.

| Prompt | Method | FDFR↑ | ISM↓ | BRISQUE↑ | FID↑ |
|---|---|---|---|---|---|
| "a photo of sks person in front of eiffel tower" | Anti-DB | 0.21 | 0.07 | 39.12 | 431.10 |
| | SimAC | 0.17 | 0.07 | 41.23 | 432.26 |
| | DisDiff | 0.25 | 0.06 | 39.98 | 439.76 |
| | MAAT | **0.68** | **0.05** | **43.63** | **499.37** |
| "a photo of sks person looking at the mirror" | Anti-DB | 0.30 | 0.16 | 39.27 | 464.13 |
| | SimAC | 0.25 | 0.28 | 40.07 | 464.47 |
| | DisDiff | 0.50 | 0.14 | 38.88 | 474.23 |
| | MAAT | **0.93** | **0.07** | **46.35** | **479.23** |
| "a photo of sks person sitting on the chair" | Anti-DB | 0.43 | 0.08 | 42.98 | 393.29 |
| | SimAC | 0.38 | 0.15 | 41.21 | 318.27 |
| | DisDiff | 0.81 | 0.03 | 46.18 | 457.22 |
| | MAAT | **0.98** | **0.01** | **49.04** | **516.93** |

Prompt 2 – "sks person personifying resilience: as a lone human figure standing knee-deep in churning ocean waves during a storm, they defiantly raise their fists toward the darkened sky while soaked in seawater and rain, rendered in dramatic cinematic lighting."

Prompt 3 – "A photorealistic portrait of sks person holding a Maltese puppy, city background, best quality, ultra-high resolution."

Table 16: Performance of MAAT under three diverse prompts on the VGGFace2 dataset.

| Prompt | FDFR↑ | ISM↓ | BRISQUE↑ | FID↑ |
|---|---|---|---|---|
| Prompt 1 | 0.76 | 0.06 | 32.88 | 432.14 |
| Prompt 2 | 0.92 | 0.04 | 31.16 | 454.53 |
| Prompt 3 | 0.77 | 0.11 | 36.00 | 410.94 |

The results are shown in Table 16. Across all three prompts, MAAT maintains competitively low ISM values, confirming its attack effectiveness and demonstrating strong robustness to prompt diversity and complexity.

**More Cross-Model Evaluations.** In the main text, we perform a cross-model analysis of the anti-customization effects of SD v1.4, SD v1.5 and SD v2.1 models. Here, we conducted additional experimental evaluations on the VGGFace2 dataset using SD2.0 and Realistic Vision V2.0. The results in Table 17 and Table 18 indicate that the MAAT method continues to show significant improvements on various diffusion versions, demonstrating the generalization of our approach.

Table 17: Evaluation results under SD2.1 (training) and SD2.0 (testing) model version mismatch on VGGFace2 dataset.

| Method | FDFR↑ | ISM↓ | BRISQUE↑ | FID↑ |
|---|---|---|---|---|
| Anti-DB | 0.22 | 0.39 | 44.94 | 316.25 |
| SimAC | 0.68 | 0.32 | 43.11 | 351.10 |
| DisDiff | 0.82 | 0.22 | 44.33 | 301.06 |
| MAAT | **0.83** | **0.19** | **45.84** | **358.94** |

## G.5 CLUSTERING DETAILS OF ANTGE

**Computational Complexity.** We do not perform the clustering as 9 independently for each image due to the substantial computational cost involved. Specifically, computing the CAI values for even a single image requires extensive forward and backward passes. Concretely, calculating

Table 18: Evaluation results under SD2.1 (training) and Realistic Vision V2.0 (testing) model version mismatch on VGGFace2 dataset.

| Method | FDFR↑ | ISM↓ | BRISQUE↑ | FID↑ |
|--------|-------|------|----------|------|
| Anti-DB | 0.79 | 0.13 | 37.47 | 482.96 |
| SimAC | 0.88 | 0.19 | 41.16 | 471.59 |
| DisDiff | 0.83 | 0.25 | 40.91 | 453.15 |
| MAAT | **0.96** | **0.09** | **41.20** | **513.01** |

$CAI(t, t')$ for one image involves 99 base timesteps ($t \in 10, 20, ..., 990$), each compared against 999 timesteps. This entails 99 × (6 adversarial attacks × (1 forward + 1 backward) + 999 forward passes), where one backward pass is approximately equivalent to 3 forward passes. As a result, the total computational cost amounts to roughly 101,277 forward passes per image. Performing this computation for every individual image would be impractical. To manage this complexity, we instead calculate the CAI values averaged across 500 randomly sampled images from CelebA-HQ as C, and perform clustering only once based on this averaged CAI distribution. The obtained clustering centroids are then consistently applied across all images and datasets. This approach significantly reduces complexity and achieves good generalization, as evidenced by the substantial improvement in results on the VGGFace2 dataset.

**Result of Clustering.**  In Table 19, we present the exact resulting cluster centroids (timesteps) obtained from our clustering. These cluster centers effectively cover both the sensitive stages (early) and the stable stages (later) of the diffusion process. We further observed that when performing clustering individually on each image, the resulting cluster centers exhibit only slight variations, while the overall trend remains consistent—early timesteps are divided into more clusters, whereas later timesteps are smoother and grouped into fewer clusters (2–3). This demonstrates the stability and consistency of the proposed clustering strategy.

Table 19: Result of Clustering.

| Cluster Index | 1 | 2 | 3 | 4 | 5 | 6 | 7 | 8 |
|---------------|-----|-----|-----|-----|-----|-----|-----|-----|
| centroids | 20 | 60 | 90 | 130 | 180 | 230 | 510 | 850 |

# H  TIME COMPLEXITY OF MAAT

Our method involves two steps per iteration:

Surrogate model updates: The surrogate model is updated three times, each requiring one forward and one backward pass, resulting in a complexity of 3 × (1 forward + 1 backward).

Adversarial sample updates: The adversarial samples undergo six updates per iteration, each update involving eight gradient computations. This totals 6 × 8 × (1 forward + 1 backward) per iteration. The total complexity per iteration is 51×(1 forward + 1 backward), for 30 iterations in total. Since one backward pass is roughly equivalent to three forward passes, the overall time complexity is approximately 6,120 forward passes.

For comparison, the Anti-DB method consists of three steps per iteration: Each step includes three surrogate model updates and six adversarial sample updates, but each adversarial update requires only one gradient computation, yielding a total complexity of 18×(1 forward + 1 backward) per iteration. With 50 iterations in total, the overall time complexity is about 3,600 forward passes. Therefore, as shown in Table 5, MAAT requires roughly 1.7 times more computation time than Anti-DB.

DisDiff and SimAC further add a search for the optimal timestep for each image based on Anti-DB, which further increases the computational overhead. As a result, the runtime of MAAT in Table 5 is comparable to that of SimAC and DisDiff.

In summary, while MAAT has higher per-iteration complexity, its reduced number of iterations keeps the overall computational cost within a reasonable range.

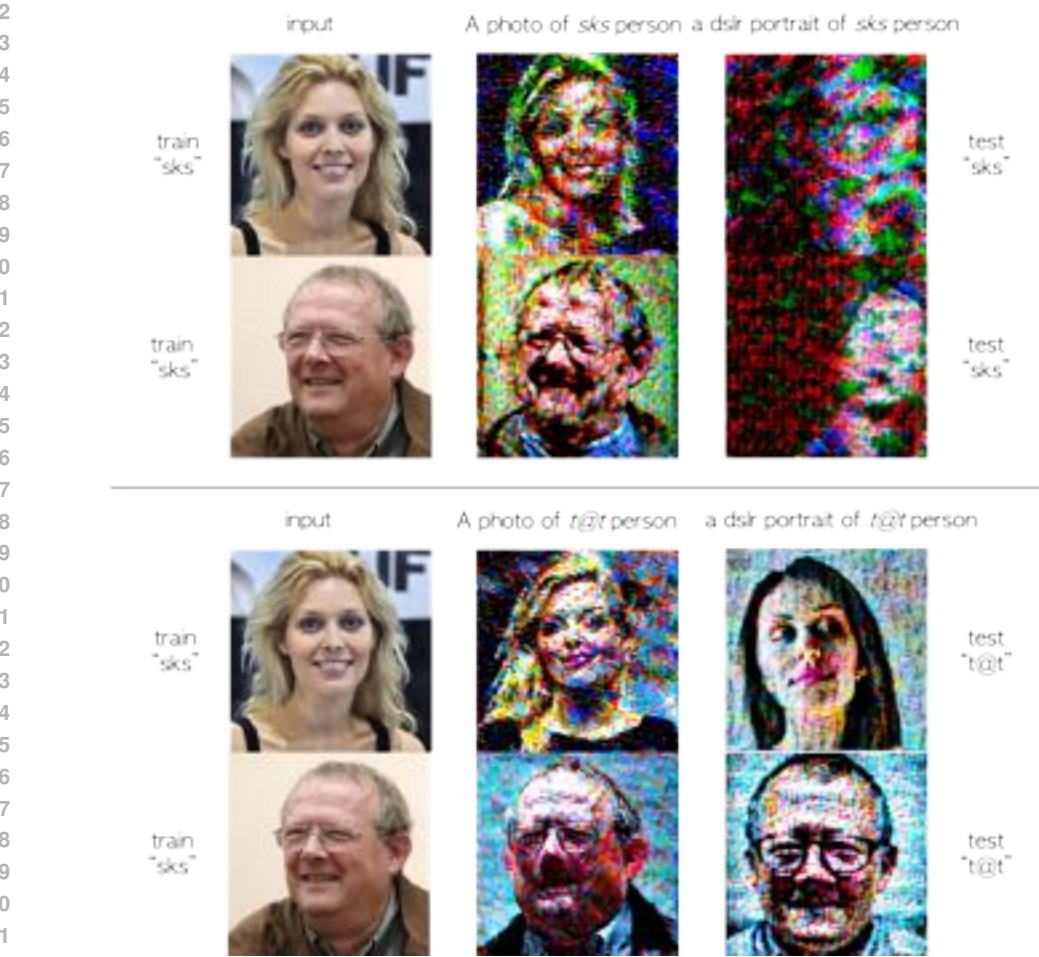

Figure 7: Quantitative results of prompt mismatch under two prompts on VGGFace2 dataset. The training rare identifier [v] is "sks" and the customization test rare identifier is "t@t". This aims to test performance when the prompt used for customization isn't foreseen.

# I   LIMITATIONS AND FUTURE WORK

While this study provides a comprehensive analysis of adversarial attacks against optimization-based personalized generation methods (e.g., DreamBooth, Textual Inversion), its scope is inherently limited to scenarios requiring model fine-tuning. The attack mechanisms explored here primarily target the training process disruption by injecting adversarial perturbations during parameter updates. However, recent advancements in training-free personalized generation (e.g., InstantBooth, PhotoMaker) leverage conditional control mechanisms without model fine-tuning, which fundamentally alters the threat landscape.

Our method is also challenged when adversaries employ purification techniques such as DiffPure to remove added perturbations. Such adversarial purification can successfully restore image quality to a level close to clean data. While this exposes a limitation of current perturbation-based defenses, it is a challenge shared by most existing approaches, including the baselines used in this paper. Developing more robust, purification-resistant defenses therefore represents a promising avenue for future research.

# J   VISUALIZATION

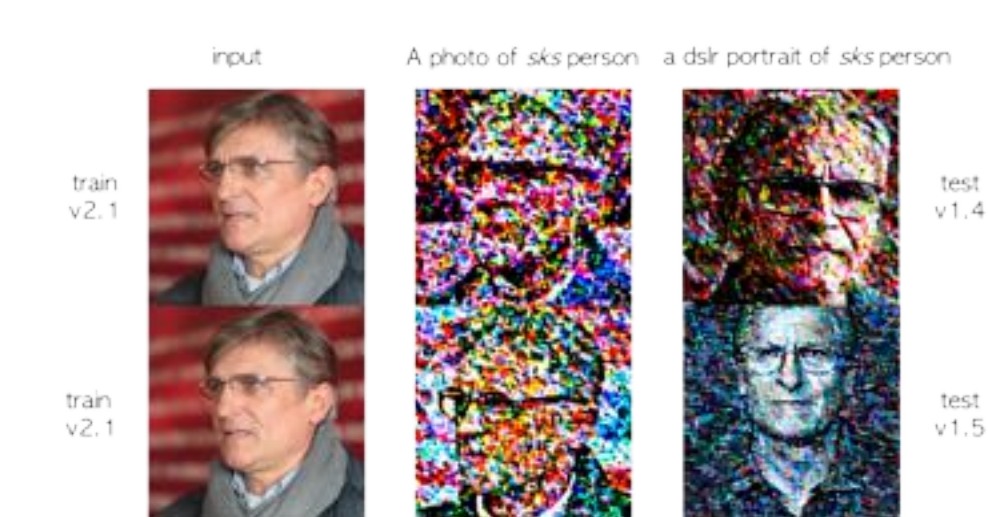

Figure 8: Quantitative results of models mismatch under four prompts on VGGFace2 dataset. The training uses stable diffusion v2.1, and the testing uses v1.4 and v2.1 in combination with the training model, respectively, to test the sensitivity of the method to the model version.

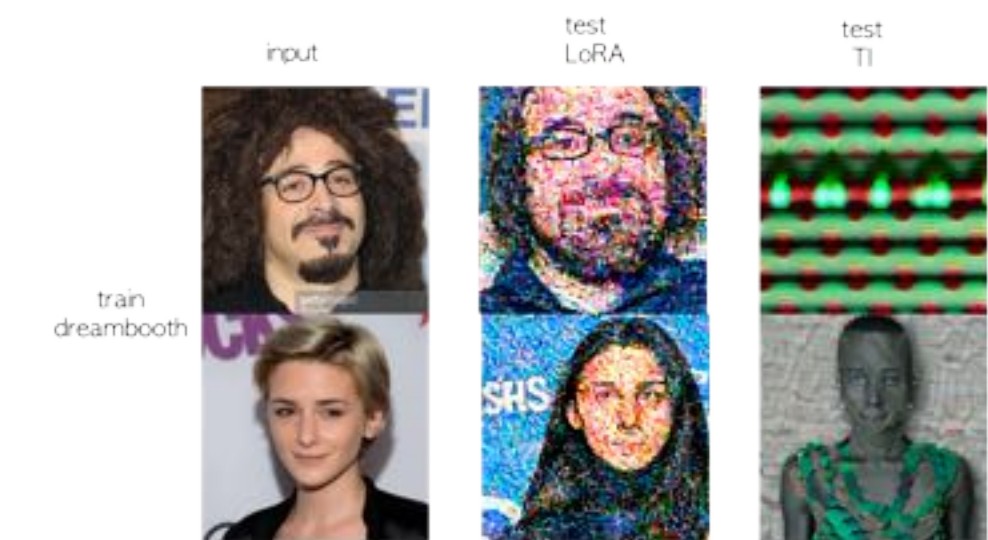

Figure 9: Quantitative results of customization mismatch on VGGFace2 dataset. The training is based on Dreambooth and the customization test is based on Lora or Textual Inversion (TI).

