# OpenReview forum: "MAAT: Multi-timestep Alternating Adversarial Training Against Personalized Content Security in Diffusion Models"
_ICLR.cc/2026/Conference — ICLR 2026 Conference Withdrawn Submission_

### Official Review · Reviewer_rDC6 · 2025-10-29

**Soundness:** 3
**Presentation:** 1
**Contribution:** 2
**Rating:** 2
**Confidence:** 3

**Summary:**

This paper introduces MAAT (Multi-timestep Alternating Adversarial Training), a new framework for protecting personal images from unauthorized fine-tuning in diffusion models such as DreamBooth and LoRA. Instead of attacking diffusion models at a single timestep, MAAT formulates the attack as a sparse multi-task optimization problem across diffusion timesteps. It introduces two main components:

1. Adaptive Non-uniform Timestep Gradient Ensemble (ANTGE): selects representative timesteps by clustering their gradient similarity and cross-timestep attack impact, improving attack efficiency and transferability.
2. Layer-Aware Attention Targeting (LAT) loss: selectively disrupts both self-attention and cross-attention layers that most strongly correlate with identity-specific regions.

MAAT also employs a two-stage alternating training paradigm where the surrogate diffusion model and adversarial noise co-evolve. Experiments on **CelebA-HQ** and **VGGFace2** show that MAAT surpasses prior anti-personalization methods such as Anti-DreamBooth, SimAC, DisDiff, and CAAT in both white-box and black-box scenarios. The method improves Face Detection Failure Rate (FDFR) by up to 20% and reduces Identity Similarity Matching (ISM) by 6.5%, while maintaining comparable image quality.

**Strengths:**

- Proposes a theoretically grounded multi-timestep optimization framework for adversarial attacks on diffusion models.
- Comprehensive experiments on two benchmark datasets, covering white-box, black-box, and cross-model settings.
- Provides ablation and theoretical analysis that link timestep sensitivity, gradient alignment, and empirical performance, although in Appendix.

**Weaknesses:**

See questions

**Questions:**

- Can the authors compare with more advanced baselines in this line of work, such as Mist v2, MetaCloak, or Glaze? The paper already cites MetaCloak in the related work but does not include it in the experimental comparison. Including these stronger baselines would help position MAAT more clearly within the current landscape of anti-personalization and protection methods.

- Can the authors specify the perturbation bound used for the L∞ PGD attack? Is the same bound applied to all baseline methods? Furthermore, how does varying the perturbation bound affect attack performance and visual quality? A sensitivity analysis on this parameter would strengthen the empirical section.

- ALL images in the paper appear unprofessionally blurry and of low visual quality, making difficult to assess the comparative effectiveness of different methods. It also makes readers hard to understand. If higher-quality visualizations cannot be provided during rebuttal (for instance, due to submission format constraints), I would recommend rejection since it significantly hinders evaluation.

Mistv2: https://github.com/psyker-team/mist-v2

MetaCloak: https://github.com/liuyixin-louis/MetaCloak

Glaze: https://arxiv.org/abs/2302.04222

---

### Official Review · Reviewer_JSwj · 2025-10-31

**Soundness:** 3
**Presentation:** 1
**Contribution:** 3
**Rating:** 4
**Confidence:** 4

**Summary:**

This paper proposes MAAT (Multi-timestep Alternating Adversarial Training), a new adversarial attack framework against unauthorized personalization of diffusion models. MAAT formulates adversarial attacks for diffusion models as a sparse multi-task optimization problem across diffusion timesteps and introduces two key techniques: 1) Adaptive Non-uniform Timestep Gradient Ensemble (ANTGE) for efficient and effective timestep selection. 2) Layer-Aware Attention Targeting (LAT) loss to selectively disrupt identity-relevant self- and cross-attention layers. Experiments on VGGFace2 and CelebA-HQ datasets show large improvements compared to existing defenses like Anti-DreamBooth, SimAC, and DisDiff, under both white-box and black-box settings.

**Strengths:**

1: Novelty: The paper reformulates diffusion attacks as sparse multi-task optimization and proposes methods to select the most relevant timestep and layers for adversarial attacks. In addition, it proposes an alternating adversarial training strategy that optimizes the surrogate models and adversarial images at the same time.
2. Comprehensive evaluation: The authors test across multiple datasets, prompts, model versions, and fine-tuning methods (DreamBooth, LoRA, TI), with solid improvements over prior work.

**Weaknesses:**

1. The figures are too blurry to see, which hinders the readability of the paper.
2. In the ablation study G.3, the LRTGE baseline uses a larger number of groups compared to ANTGE. The ablation study should use the same number of groups for a fair comparison.
3. It is unclear how the texture-critical layers are selected in Eq. (11).
4. In the black-box experiment, the "model mismatch" case only considers the stable diffusion family. How about the transferability across model families and architectures?
5. The image quality after the adversarial attacks should also be evaluated to make sure the image looks natural.

**Questions:**

1. Please pay attention to the mismatched quote marks throughout the paper. For example, Line 378 ”a photo of sks person” ”a dslr portrait of sks person”.
2. Others: see the weakness section.

---

### Official Review · Reviewer_y834 · 2025-11-01

**Soundness:** 2
**Presentation:** 1
**Contribution:** 1
**Rating:** 2
**Confidence:** 3

**Summary:**

This paper proposes MAAT, an adversarial defense framework against unauthorized personalization of text-to-image diffusion models (e.g., DreamBooth). The core idea is to protect user-uploaded images by embedding imperceptible perturbations that disrupt the fine-tuning process used in Personalized Content Synthesis (PCS). MAAT introduces three main technical components: Adaptive Non-uniform Timestep Gradient Ensemble, Layer-Aware Attention Targeting loss, and Two-stage Alternating Adversarial Training. Experiments on VGGFace2 and CelebA-HQ show MAAT outperforms prior methods.

**Strengths:**

1. The paper correctly identifies key limitations of prior works: (a) single-timestep or uniform timestep selection; (b) uniform attention disruption without layer-wise relevance modeling; and (c) surrogate model misalignment in ASPL.

2. The formulation of adversarial attack as a sparse multi-task optimization problem is principled. Lemma 2 provides a theoretical justification for clustering timesteps based on gradient similarity, with a controllable Pareto-distance bound.

3. The paper evaluates across white-box/black-box settings, including prompt mismatch, model version mismatch (SD v1.4/1.5/2.1), and personalization method mismatch (DreamBooth → LoRA/TI). Results are consistent and statistically significant.

4. Despite computing gradients at 8 timesteps, MAAT converges faster than baselines (Table 5), showing that sparse but adaptive timestep selection is more effective than dense or greedy strategies.

**Weaknesses:**

1. All the figures in the paper are in very poor quality, and not readable. Therefore, the paper do not fully align with the guidelines of the ICLR.

2. MAAT’s contributions are heavily built upon recent works:The two-stage training is a refinement of ASPL from Anti-DreamBooth. Attention-based disruption follows DisDiff and CAAT. Multi-timestep attack ideas appear in DADiff (“local random-timestep gradient ensemble”) and SimAC (gradient-magnitude-based selection). The paper positions itself as a unifying framework, but the novelty of each component is moderate.

3. MAAT only addresses fine-tuning-based PCS (DreamBooth, LoRA, TI). It explicitly acknowledges failure against training-free methods like InstantID or PhotoMaker, which are increasingly popular. This limits real-world applicability.

4. Table 14 shows JPEG compression (Q=30) or strong Gaussian blur (K=7) nearly nullifies MAAT’s protection (FDFR drops to 0.06–0.07). Real-world image sharing almost always involves such preprocessing, raising practical concerns.

**Questions:**

1. The LAT loss relies on SAM-generated facial masks. How would MAAT adapt to non-facial subjects (e.g., pets, logos) where identity-critical regions are less defined?

2. DADiff also uses a dual-stage attack and attention disruption. How does MAAT’s ANTGE differ fundamentally from DADiff’s “local random-timestep gradient ensemble”? Is the gain primarily from non-uniform clustering?

3. The paper mentions DiffPure as a threat. Have you tested MAAT against diffusion-based purification or JPEG-aware adversarial training?

4. MAAT uses 30 epochs vs. Anti-DB’s 50 (Appendix H), yet Table 5 reports total time. Could the performance gain be partly due to more aggressive optimization rather than architectural novelty?

**Details Of Ethics Concerns:**

Please check the figures. all the figures are of low quality. May be, the authors tried to compress the PDF, however, the content is not readable.

---

### Note · Authors · 2025-11-13

I have read and agree with the venue's withdrawal policy on behalf of myself and my co-authors.